# Monitoring of Hydrological Resources in Surface Water Change by Satellite Altimetry

**Wei Li** [1,2,3,4,*] **, Xukang Xie** [1,3] **, Wanqiu Li** [5] **, Mark van der Meijde** [2] **, Haowen Yan** [1,3] **, Yutong Huang** [1,3] **, Xiaotong Li** [1,3] **and Qianwen Wang** [1,3]

1. Faculty of Geomatics, Lanzhou Jiaotong University, Lanzhou 730070, China
2. Department of Earth Systems Analysis (ESA), Faculty of Geo-Information Science and Earth Observation (ITC), University of Twente, 7514 AE Enschede, The Netherlands
3. National-Local Joint Engineering Research Center of Technologies and Applications for National Geographic State Monitoring, Lanzhou 730070, China
4. School of Civil Engineering, Hexi University, Zhangye 734000, China
5. School of Surveying and Geo-Informatics, Shandong Jianzhu University, Jinan 250101, China
* Correspondence: geosci.wli@lzjtu.edu.cn; Tel.: +86-139-1996-9611

**Abstract:** Satellite altimetry technology has unparalleled advantages in the monitoring of hydrological resources. After decades of development, satellite altimetry technology has achieved a perfect integration from the geometric research of geodesy to the natural resource monitoring research. Satellite altimetry technology has shown great potential, whether solid or liquid. In general, this paper systematically reviews the development of satellite altimetry technology, especially in terms of data availability and program practicability, and proposes a multi-source altimetry data fusion method based on deep learning. Secondly, in view of the development prospects of satellite altimetry technology, the challenges and opportunities in the monitoring application and expansion of surface water changes are sorted out. Among them, the limitations of the data and the redundancy of the program are emphasized. Finally, the fusion scheme of altimetry technology and deep learning proposed in this paper is presented. It is hoped that it can provide effective technical support for the monitoring and application research of hydrological resources.

**Keywords:** satellite altimetry; hydrological resources; deep learning; improvement of fusion algorithm



## 1. Introduction

Water is one of our most precious natural resources [1–3]. Hydrology is the study of water, which subdivides into surface water hydrology, groundwater hydrology (hydrogeology), and marine hydrology (Figure 1). Hydrological research briefly has the following branches: Groundwater hydrology (hydrogeology) considers quantifying groundwater flow and solute transport [4]. Hydroinformatics is the adaptation of information technology to hydrology and water resources applications. Surface water flow can include flow both in recognizable river channels and otherwise. Methods for measuring flow once the water has reached a river include the stream gauge and tracer techniques. Drainage basin management covers water storage, in the form of reservoirs, and floods protection. Hydrological research can inform environmental engineering, policy, and planning. Using various analytical methods and scientific techniques, we can collect and analyze data to help solve water-related problems such as environmental preservation, natural disasters, and water management [5,6]. Hydrological resource problems are also the concern of scientists, specialists in applied mathematics and computer science, and engineers in several fields.

Geodesy, composed of various observation techniques of the earth's shape, rotation, and gravity field (and their respective temporal variations), has been playing an important role in sensing meteorological, climatological, and hydrological events. For

example, satellite gravimetry, represented by the Gravity Recovery and Climate Experiment (GRACE) and its follow-up mission (GRACE-FO), has been a unique means to monitor the distribution and redistribution of mass transport within the earth system and subsystems (e.g., hydrosphere).

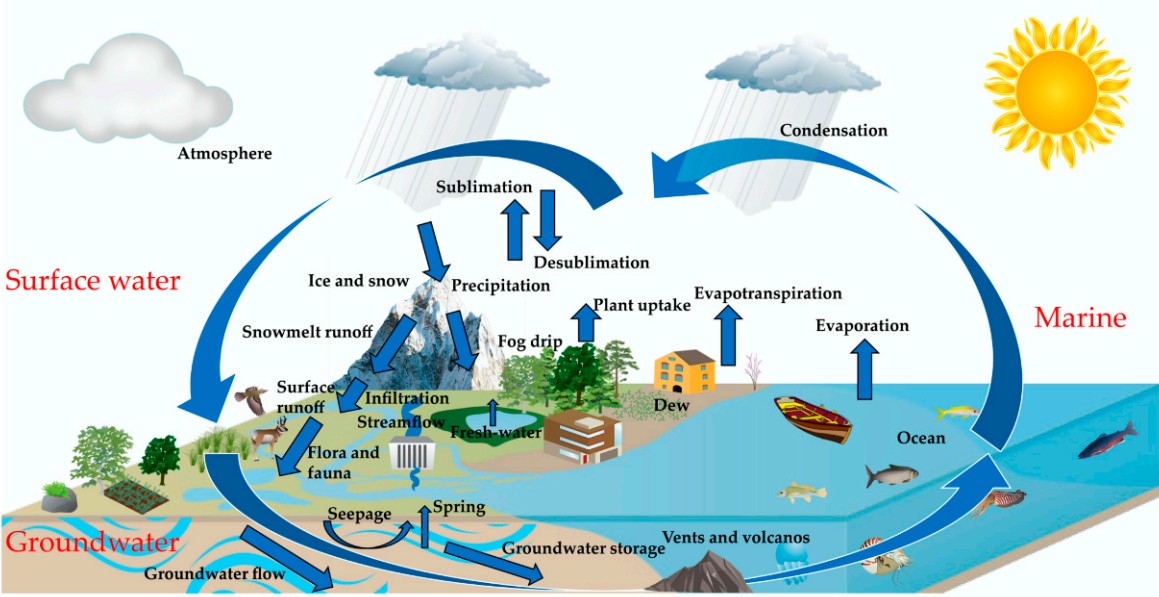

**Figure 1.** Schematic diagram of the global hydrological cycle and water resources.

Remote sensing of hydrologic processes can provide information on locations where in situ sensors may be unavailable or sparse. It also enables observations over large spatial extents. Many of the variables constituting the terrestrial water balance, for example, surface water storage [7,8], soil moisture, precipitation [9], evapotranspiration [10], and snow and ice [11–21], are measurable using remote sensing at various spatiotemporal resolutions and accuracies [7]. Sources of remote sensing include land-based sensors, airborne sensors, and satellite sensors, which can capture microwave, thermal and near-infrared data or use lidar [22]. Satellite altimetry belongs to a technology in remote sensing, which was first proposed by Kaula at the Solid Earth and Ocean Physics Congress in 1969 [23]. Its development can be divided into three stages:

(1) Stage of the experiment: The National Aeronautics and Space Administration (NASA) launched the Skylab space station with the radar altimeter S-193 in 1973. Subsequently, altimetry satellites of various countries have also entered space one after another, and a new era of satellite altimetry has begun [24].

(2) Stage of the development: Defined from the TOPEX/Poseidon mission in 1992 [25]. The satellite mission enabled the computation of ionospheric delay corrections by introducing a second altimeter frequency (C-band, 5.3 GHz) and a third frequency for the microwave radiometer (18 GHz). At the same time, the influence of wind speed on the measurement is eliminated. In this way, it has revolutionized satellite altimetry technology [26].

(3) Stage of the future: To be able to monitor land water, the research institute plans to carry out the Surface Water Ocean Topography (SWOT) (https://swot.jpl.nasa.gov (accessed on 1 June 2022)) Mission. SWOT is the world's first satellite for the global survey of the earth's surface water. (https://www.aerospace-technology.com/projects/surface-water-and-ocean-topography-swot-satellite/ (accessed on 1 June 2022)).

Table 1 gives more research information on satellite altimetry. Figure 2 shows the evolution of satellite altimetry.

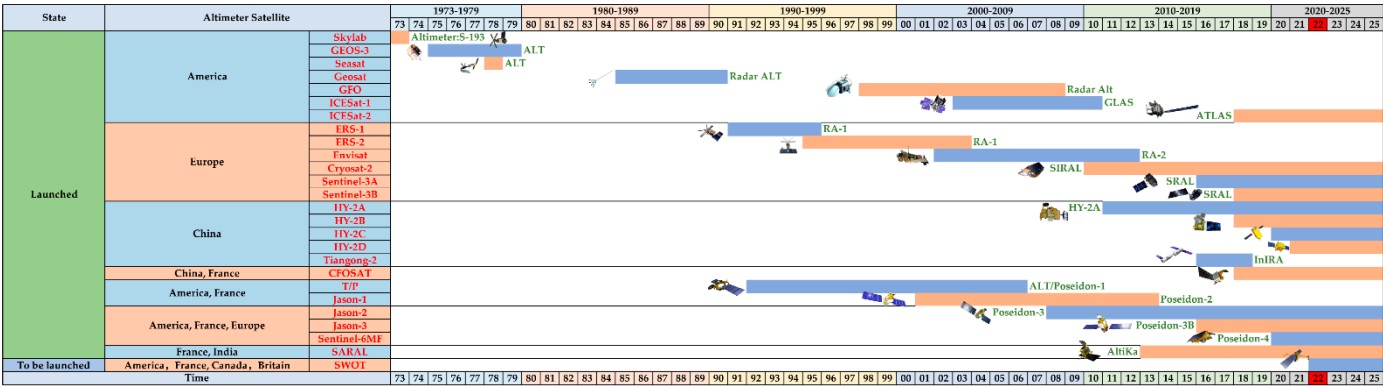

**Figure 2.** Diagram of the development process of altimetry satellites.

**Table 1.** The development process of satellite altimetry.

|  | Satellite | Agency | Service Period | Frequency Used (Band) | Repetitivity (Days) | Inclination (°) |
|---|---|---|---|---|---|---|
| **Past missions** | Skylab [27] | NASA | 1973 | Ku | — | 50 |
|  | GEOS-3 [28] | NASA | 1975–1979 | Ku | 23 | 115 |
|  | Seasat [29] | NASA | 1978 | Ku | 17 | 108 |
|  | Geosat [30] | U.S.Navy | 1985–1990 | Ku | 17 | 108.1 |
|  | ERS-1 [31] | ESA | 1991–2000 | Ku | 35 | 98.52 |
|  | T/P [32] | NASA/CNES | 1992–2006 | Ku and C | 10 | 66 |
|  | ERS-2 [31] | ESA | 1995–2011 | Ku | 35 | 98.52 |
|  | GFO [33] | U.S.Navy /NOAA | 1998–2008 | Ku | 17 | 108 |
|  | Jason-1 [34] | CNES/NASA | 2001–2013 | Ku and C | 10 | 66 |
|  | Envisat [35] | ESA | 2002–2012 | Ku and S | 35 | 98.55 |
|  | ICESat-1 [36] | NASA | 2003–2010 | 1064 nm and 532 nm | 183 | 94 |
|  | Jason-2 [37] | CNES/NASA/ Eumetsat/NOAA | 2008–2019 | Ku and C | 10 | 66 |
|  | HY-2A [38] | CAST | 2011–2020 | Ku and C | 14 | 99.34 |
|  | Tiangong-2 [39] | CAST | 2016–2018 | Ku | — | 42 |
| **Current missions** | Cryosat-2 [33] | ESA | 2010–now | Ku | 369 | 92 |
|  | Saral [40] | ISRO/CNES | 2013–now | Ka | 35 | 98.55 |
|  | Jason-3 [41] | CNES/NASA /Eumetsat/NOAA | 2016–now | Ku and C | 10 | 66 |
|  | Sentinel-3A [41] | ESA | 2016–now | Ku and C | 27 | 98.65 |
|  | ICESat-2 [36] | NASA | 2018–now | 532 nm | 91 | 92 |
|  | Sentinel-3B [42] | ESA | 2018–now | Ku and C | 27 | 98.65 |
|  | CFOSAT [43] | CNSA/CNES | 2018–now | Ku | — | 90 |
|  | HY-2B [44] | CAST | 2018–now | Ku and C | 14 and 168 | 99.34 |
|  | HY-2C [45] | CAST | 2020–now | Ku and C | 10 | 66 |
|  | HY-2D [46,47] | CAST | 2021–now | Ku and C | 10 | 66 |
|  | Sentinel-6MF [48] | ESA/Eumetsat/EU/ CNES/NOAA/ NASA | 2020–now | Ku and C | 10 | 66 |
| **Future missions** | SWOT [49] | CNES/NASA/ CSA/UKSA | 5 December 2022 [50] | Ka | 21 | 77.6 |

## 2. Application and Development

### *2.1. Applications in Hydrological Resources Monitoring*

2.1.1. Hydrological in Solid Form

Numerous studies have shown that the research field of satellite altimetry has been continuously expanded [51]. Satellite altimetry has long been not limited to traditional oceanography and geodesy, and has been widely used in terrestrial hydrology. After 50 years of development, the accuracy of measurement has been significantly improved, and the spatiotemporal resolution has also been continuously reduced.

For nearly 30 years, satellite altimeters have provided important information for understanding oceanic and inland hydrodynamics. Based on the principle of semantic association, we analyze and discover the existing research results of satellite altimetry. Research keywords, research, and application of satellite altimetry are shown in Figure 3. Various important parameters can be inferred from altimeter measurements, including sea surface height, sea surface wind speed, significant wave height, and the topography of the land, sea ice, and ice sheets. Using these parameters and long-term records of altimeter data spanning decades, a wide variety of societal applications can be realized [52–54]. During this period, satellite radar altimeter data has been successfully applied to fish tracking [55], the mass balance of ice sheet [12,19–21,56–58], estimate snow variation [14,59], lake ice thickness monitoring [60,61], severe storm forecasting [62], oil spill response [63], ship route tracking [64], iceberg monitoring [65], marine wildlife habitat monitoring [66], wetland dynamics monitoring [67], reservoir lake monitoring [68,69], flood forecasting [70], monitoring of changes in river levels [18,71], and the development of offshore wind farms [72], among which fish tracking and the development of offshore wind farms are already commercially available.

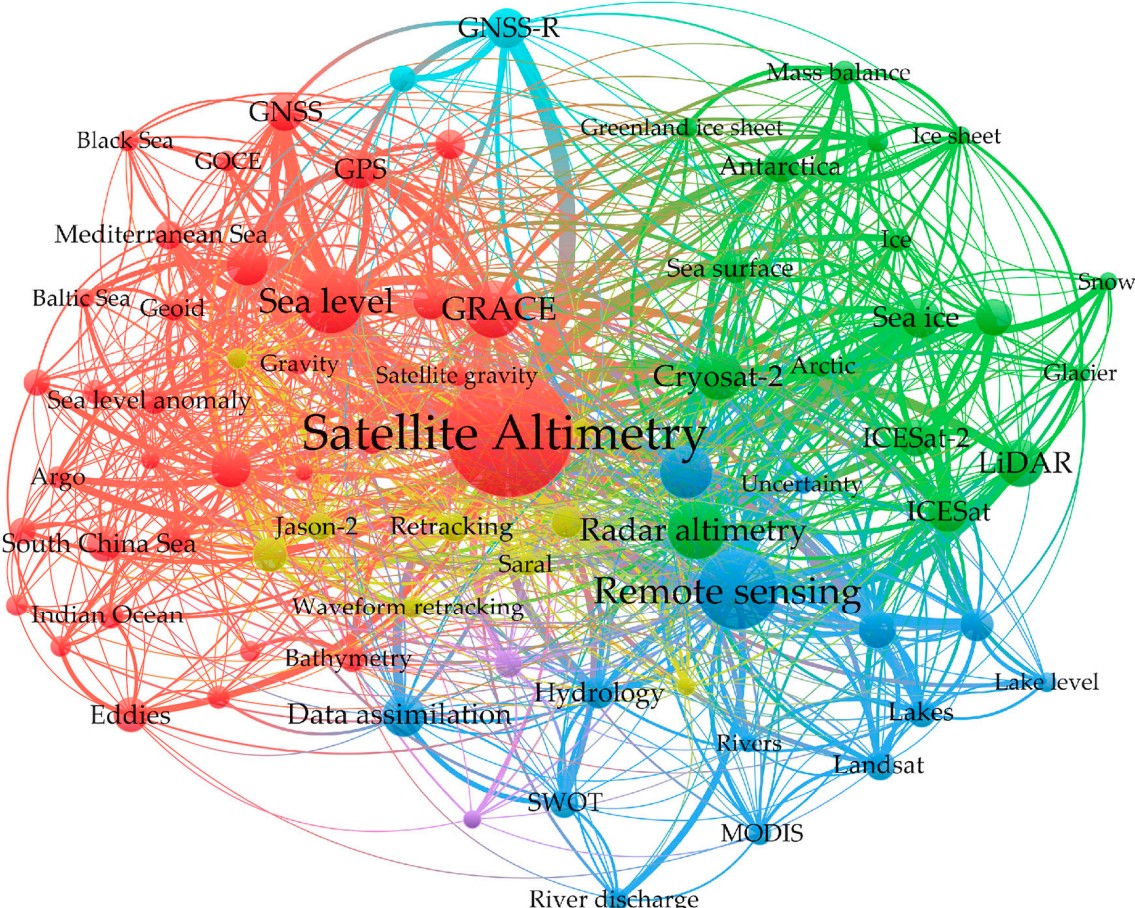

**Figure 3.** Semantic analysis of satellite altimetry and other disciplines and application fields.

2.1.2. Hydrological in Liquid Form

The application of satellite altimetry technology has expanded from initial sea-level monitoring to monitoring water level changes in inland lakes and other waters [73–81]. From the use of lake surface elevation data observed by Seasat satellites for mapping [82], the use of satellite altimetry technology to observe inland lake waters has gradually attracted attention. In the later research progress, researchers began to use altimetry satellite data to observe inland lakes, for example: using Geosat altimetry data to monitor the water level changes of several lakes such as Lake Ontario Lake and Lake Superior [83]; using T/P measurement, Gao Data studied the relationship between water level changes and precipitation in six lakes such as Lake Michigan and Lake Huron [84], studied the changes of the Indian Ocean climate change and the influence of East African lakes [85], and studied African countries affected by the Indian Ocean climate. The water level changes of lakes [86] and the water level of Lake Isabal were dynamically monitored using Envisat altimetry data, the relationship with local climate changes was analyzed [87], and the lake surface elevation changes in the northeastern Tibetan Plateau were analyzed.

The study of inland lakes and reservoirs has received increasing attention due to the increasing population, increasing water demands, and changing climate. However, in unmeasured areas, there will be limited, outdated, or nonexistent hydrological data, and there will be a lack of data-sharing mechanisms. These problems can hinder the monitoring of inland lakes and reservoirs [88]. In response to these problems, the use of satellite altimetry technology in reservoir operations and river system modeling has been successfully carried out in operational environmental monitoring of transboundary basins, such as the Yangtze and Yellow River Basins [89,90], the Amazon Basin [91], the Mekong River Basin [92], the Ganges-Brahmaputra in Southeast Asia Rivers-Mekhna Valley [68], the Indus Valley [17,18,77,93–95], the Nile Valley [96–99], and other. The above studies conclude that the results of the studies based on altimetry satellites are highly consistent with the observed data in most watersheds.

*2.2. Developments in Altimetry Data and Processing*

2.2.1. Availability of Data

We know that the satellite is observed along the orbit, so there is no observation data in the interval area between orbits, which makes the altimetry data different from the remote sensing image data. It does not completely cover the ground but presents a "fishnet-like" coverage (Figure 4). In this regard, a wide-width satellite altimeter can be developed, which refers to the combined use of a finite pulse radar altimeter and an interferometric altimeter to solve the problem of satellite coverage.

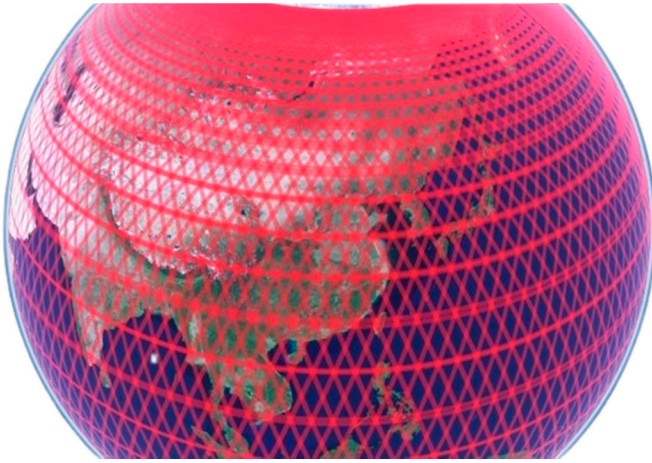

**Figure 4.** Illustration of fishnet coverage of altimetry satellite orbits.

In terms of data availability, there are also many successful research cases of teams, for example, the Hydroweb database (http://hydroweb.theia-land.fr/ (accessed on 5 June 2022)) developed by the French LEGOS laboratory based on six radar altimetry missions such as GFO and Jason-3, mainly including large rivers, lakes, and water level time series of wetlands. At present, it has more than ten years of business data and is still growing. The G-REALM database (https://ipad.fas.usda.gov/cropexplorer/global_reservoir (accessed on 5 June 2022)) for global reservoir and lake monitoring was developed by the USDA Foreign Agricultural Service, NASA, and the University of Maryland, according to Jason A series of satellites are periodically detected, and the water level information of global reservoirs and lakes can be regularly monitored every ten days. A satellite-based operating system, developed by the SASWE research group at the University of Washington, can also be used to monitor lakes and reservoirs in areas with no data. ISRO has used radar altimeter data to retrieve water level data for more than 20 years and has successfully built many applications that are of great help in assessing and managing water resources in India (www.isro.gov.in/ (accessed on 5 June 2022)). Some studies have pointed out that regional lake water level changes with different spatiotemporal resolutions. The trend of using altimeter data such as Jason-2 to detect and GRACE results is consistent, such as in the study of lake water levels on the Tibetan Plateau [100]. For the problem of land pollution in inland waters, if we want to improve the observation accuracy of altimetry data in inland waters, we can use the waveform retracking algorithm of inland lakes based on waveform purification technology [101–105].

Therefore, we have listed several data access options here. During data processing, data can be downloaded directly from any NSIDC dataset, and related information is also available in Table 2. In addition, when operating the IceBridge Portal, it will allow you to discover, filter, and access IceBridge data. In addition to these single dataset and mission options, three missions are also accessible through the NASA Earthdata Search as well as the Application Programming Interface (API). The following information provides detailed descriptions and examples of these data access options: (1) Earthdata Search: These can help us through the process of exploring and accessing ICESat, IceBridge, and ICESat-2 data that overlap in the area and/or time frame of interest in the NASA Earthdata Search application. It also includes subset and reformatting options for ICESat and ICESat-2 data (https://search.earthdata.nasa.gov/search (accessed on 5 June 2022)). (2) API: These can programmatically access satellite altimetry data through application programming interfaces or APIs based on spatial and temporal filters. The resulting data is returned as a single file or a multi-file zip, depending on the amount of data granularity requested (https://nsidc.org/data/user-resources/help-center/programmatic-data-access-guide (accessed on 5 June 2022)). (3) API ports can also be accessed through Python programs (https://github.com/nsidc/NSIDC-Data-Access-Notebook (accessed on 5 June 2022)).

**Table 2.** List of available satellite altimetry data.

| Data Access | Distributed By | Resources |
|---|---|---|
| AVISO+ [106] | AVISO, CNES, CTOH | https://www.aviso.altimetry.fr/en/home.html (accessed on 5 June 2022) |
| RADS [107,108] | NOAA, Altimetrics | http://rads.tudelft.nl/rads/rads.shtml (accessed on 5 June 2022) |
| OpenADB [109] | DGFI-TUM | https://openadb.dgfi.tum.de/en/ (accessed on 5 June 2022) |
| Aviso-CNES [106] | AVISO, CNES | https://aviso-data-center.cnes.fr/ (accessed on 5 June 2022) |
| COSDSC | NSOAS | https://osdds.nsoas.org.cn/ (accessed on 5 June 2022) |
| CTOH [106] | CTOH | http://ctoh.legos.obs-mip.fr/ (accessed on 5 June 2022) |

At the same time, we find that if the area of the lake is small, the footprint of the altimetry satellite is also not passed. This will lead to the lack of corresponding altimetry data, and the information on lake water level monitoring cannot be obtained. Although the current solution is to develop wide-format altimetry satellites, there are currently very few altimeter satellites equipped with wide-format altimeters. For example, the wide-format altimeter on Tiangong-2 has stopped running. The current situation is that there are few satellites, a short operating time, and relatively little data. Therefore, the solution for developing a wide-format altimeter needs to continue to be improved or find other solutions. Fortunately, we know that the monitoring data of the hydrological station has the advantages of high precision and continuity, and it can be used as the verification information of the satellite altimetry data (Table 3).

**Table 3.** Hydrological database using satellite altimetry data.

| Products | Resources | Applications |
|---|---|---|
| River and Lake [110] | http://altimetry.esa.int/riverlake/shared/main.html (accessed on 6 June 2022) | Rivers, lakes, and reservoirs monitoring |
| DAHITI [111] | https://dahiti.dgfi.tum.de/en/ (accessed on 6 June 2022) | Rivers and lakes monitoring |
| Hydroweb [112] | http://hydroweb.theia-land.fr/ (accessed on 6 June 2022) | Lakes and reservoirs monitoring |
| HydroSat [113] | http://hydrosat.gis.uni-stuttgart.de (accessed on 6 June 2022) | Rivers, lakes, and reservoirs monitoring |
| G-REALM [114] | https://ipad.fas.usda.gov/cropexplorer/global_reservoir/ (accessed on 6 June 2022) | Lakes and reservoirs monitoring |

### 2.2.2. Applicability of the Program

We know that monitoring inland lakes falls under the category of hydrology. Although the use of radar altimetry data has been widely carried out in the field of hydrological remote sensing research [23], one of the main challenges remains the need to demonstrate the reliability of radar altimetry data in a different area and spatiotemporal resolution. At the same time, it is difficult to choose which programs to process altimetry data and which program to use, and the applicability of the program is average.

Table 4 shows the program details, we can see that: (1) BRAT is currently one of the mainstream software for processing altimetry data. It is compatible with the RADS altimetry database and is widely used, but it lacks analysis functions such as processing time series. (2) ncBrowse is a Java application that can view altimetry data in netCDF format, but it lacks functions for processing and analyzing altimetry data. (3) Panoply can visualize the altimetry data in netCDF format, but it lacks the function of processing and analyzing altimetry data. (4) NCO (netCDF Operator) is a set of programs for processing netCDF files, but it cannot analyze and visualize altimetry data. (5) ATSAT is a MATLAB-based software package that can be used to process and analyze altimetry data, but the software package itself cannot acquire altimeter data, and can only process Jason-1, Jason-2, Jason-3, Saral, and Sentinel-3 five data from altimetry satellites, but also lacks altimetry data preprocessing capabilities.

**Table 4.** Programs for processing altimetry data.

| Code Access | Resources | Functions | | | |
|---|---|---|---|---|---|
| | | Preprocessing | Processing | Analysis | Visualization |
| BRAT [115] | http://www.altimetry.info/ (accessed on 6 June 2022) | × | √ | √ | √ |
| ncBrowse [116] | https://www.pmel.noaa. gov/epic/java/ncBrowse/ (accessed on 6 June 2022) | √ | × | × | √ |
| Panoply | http://www.giss.nasa.gov/ tools/panoply/ (accessed on 6 June 2022) | × | × | × | √ |
| NCO [117] | http://nco.sourceforge.net/ (accessed on 6 June 2022) | √ | × | × | × |
| ATSAT [118] | https://sglab.ut.ac.ir/ software-and-data/ (accessed on 6 June 2022) | × | √ | √ | √ |

The key point of using altimetry data in hydrology is to provide useful modeling tools [7,119–121]. In addition, a big challenge is to improve the robustness and redundancy of data processing programs, reduce parameter interaction, and improve multi-source data processing capabilities. Although many institutions also provide corresponding data processing plans and guidance information [122,123]. Based on these studies, we will also integrate machine learning and altimetry data processing. In this way, data processing programs with good completeness can be developed.

As we all know, the concept of artificial intelligence (AI) has been quite broad [51], and the introduction of artificial intelligence into the field of altimetry is still in the exploratory stage [23,124]. Deep learning, one of the branches of artificial intelligence, is currently applied to the ocean. Remote sensing and aiding in the improvement of marine elements have shown effectiveness [124–127], proving the possibility of artificial intelligence mining the potential value of altimetry data. Each altimetry task has to go through a verification and calibration phase, and calibration is required to improve accuracy. For example, linear or polynomial regression methods are usually used to correct wave height and wind. However, if the performance of the altimeter is sometimes complex, still, it is not enough to accurately calibrate or reduce errors, and deep learning is effective in solving this problem [44]. South Korea applied artificial neural networks to expand the existing groundwater monitoring network in South Korea through a large amount of hydrological observation data and played an important role in evaluating the relationship between surface water and groundwater [128].

The schematic diagram of AI in the field of altimetry can not only mine the dynamic value of static data but also show its unique advantages in improving data accuracy (Figure 5). The finite pulse radar altimeter, the traditional radar altimeter, is the most important spaceborne radar altimeter at present, with the longest application time and the largest number of satellites [24], such as the Poseidon series altimeter of Jason satellite (Figure S1). The spaceborne altimeter transmits a pulse signal to the earth and receives the return waveform, and measures the distance between the satellite and the earth's surface covered by the radar footprint by tracking the received waveform [75,128]. Currently, as more and more radar datasets are made available, there is a strong interest in using radar data as input to various deep learning algorithms [129]. In order to further improve the accuracy of satellite altimetry data, experts and scholars have conducted a lot of research, which mainly focuses on optimizing the correction parameters of geophysics and propagation media and waveform retracking technology [130–132]. When the waveform received by the radar altimeter is disturbed, the received signal will be different from the standard model, causing the distance measured by the altimeter to deviate from the actual

distance. At present, the radar industry has begun to introduce AI into radar waveform signal processing, and has verified the feasibility [133–135].

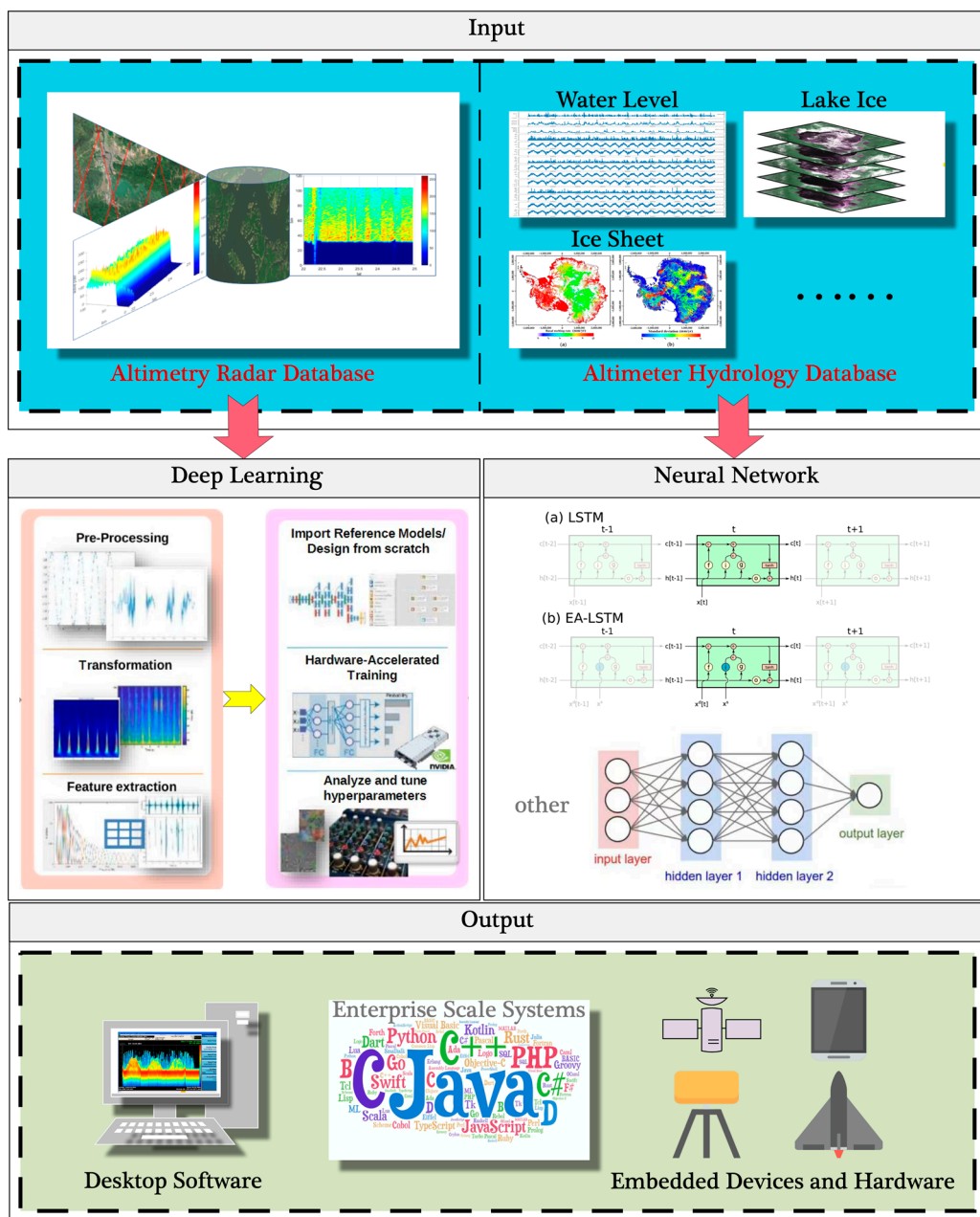

**Figure 5.** Schematic diagram of the fusion strategy of satellite altimetry and AI.

## 3. Challenge and Opportunity

### 3.1. Limitations of Data and Processing Methods

3.1.1. Spatiotemporal Resolution

For a satellite, if its orbital repetition period is shorter, the interval between adjacent orbits is larger. In this way, the repetition period and the interval between adjacent orbits are contradictory, and eventually, the spatial sampling and time sampling of satellite observations cannot obtain a higher resolution at the same time [136].

For this situation, the classic solution is to form a constellation of altimetry satellites with the same configuration to solve the problem of spatial and temporal resolution [137–140]. At present, SARIn altimeter, which is an important development direction of satellite altimetry technology in the future, will make the transition from traditional one-

dimensional, along-orbit profile altimetry to two-dimensional wide-width interferometric altimetry [24]. This will greatly improve spatiotemporal resolution. This solution is being explored gradually, and the upcoming SWOT satellite is equipped with a SARIn altimeter, which is expected to achieve good results.

With its developed wide-field mapping technology, the SWOT mission is expected to overcome some of the challenges of satellite altimetry, especially the limited spatial resolution of ground orbits and low revisit times, for example, for small-scale mesoscale observations in the range of 10–200 km. The existing altimeter cannot capture well, and the SWOT task will probably solve this challenge. This will improve our understanding of ocean dynamics. Therefore, SWOT satellites will become the key data support for studying water resources from space observations and set standards for future satellite altimetry [141,142]. In the future, the international altimetry community will strongly support the data and information products of future tasks such as SWOT and the data and information products of altimetry tasks such as Jason-3 and Sentinel-6. They are used to synergistically address the many unsolved environmental challenges facing society today, whether global or regional.

### 3.1.2. Terrain Detection Capability

At the same time, it is found that the finite pulse satellite altimeter is suitable for relatively uniform and relatively smooth surface types such as oceans and large ice sheets. Similar to inland glaciers, lakes, waters, and other complex or undulating terrains, it will lead to data loss or information distortion. In this regard, the accuracy of altimetry can be improved by improving data processing algorithms such as altimeter sensors and waveform retracking [143]. The interferometric altimeter is less affected in terrain areas with complex surface types or large fluctuations, so the accuracy is higher. Therefore, we can use the images of multiple small satellites in the distributed spaceborne InSAR system to perform multi-baseline interference, which can improve the accuracy of altimetry.

The development of altimeter data and the refinement of retracking algorithms have always relied on quality-controlled measured data from oceans and water bodies. Many successful cases of altimeter applications require measured records of rivers, reservoirs, etc., to verify and calibrate the operating system. Different retracking algorithms can also be used to process different satellite mission data in different areas. For instance, ALES is designed for Jason1/2 and Envisat data in the open ocean and coastal areas, the X-TRACK retracking algorithm is specially designed for coastal areas, ALES+ can be applied to inland waters [21,144,145], and the Goddard Space Flight Center has designed several retracking algorithms for ice areas [146]. For the Jason-2 SGDR data in the offshore area, the selection method of the optimal Gaussian low-pass filter radius can be used to determine the sea surface height [147–151].

### 3.1.3. Root Tracing Algorithm

Different from the data processing method in the public sea area, the monitoring of inland waters with altimetry data requires special processing [152–155]. The most widespread application of satellite radar altimetry is global sea-level change. In recent years, satellite altimetry technology has been applied to the monitoring of inland lakes, but land signal pollution often occurs. This greatly affects the accuracy of satellite altimetry. We can improve the corresponding accuracy by optimizing the correction parameters (for example dry and wet tropospheric correction, ionospheric correction, etc.) and improving the waveform retracking algorithm.

The existing altimeters are mainly finite pulse radar altimeters, and we can use physical and empirical algorithms to process their data. However, the important development direction of satellite altimetry technology in the future is the SARIn altimeter. It will gradually transition the traditional one-dimensional and along-section altimetry to two-dimensional and wide-area interferometric altimetry, which greatly improves the temporal and spatial resolution of data. At present, only Tiangong-2 and Cryosat-2 satellites are

equipped with SARIn altimeters, and SWOT satellites will also be equipped with SARIn altimeters in the future. The existing algorithms and models are no longer applicable for retracking processing of the complex waveform of the SAR/SARIn spaceborne radar altimeter. What is exciting is that the physical model algorithms and empirical statistical algorithms proposed by scholars will be more suitable for the data processing of the SARIn altimeter.

### 3.2. Application and Expansion of Multi-Source Data

#### 3.2.1. Prior Information on Inland Lake Monitoring

We can verify it with high-precision inversion of satellite data. For example, when using the observation data of Tiangong-2 InIRA to invert some lakes on the Tibetan Plateau, some of these lakes lack the measured water levels of hydrological stations. The accurately retrieved water level of Cryosat-2 was used as the observation data of Tiangong-2 InIRA [24]. However, this method needs to consider the running time of the satellite, the accuracy of the inversion, etc., so it still needs to be improved. However, due to the lack of hydrological stations in some inland lakes and the shortage of data resources, real-time sharing needs to be improved (Figure 6).

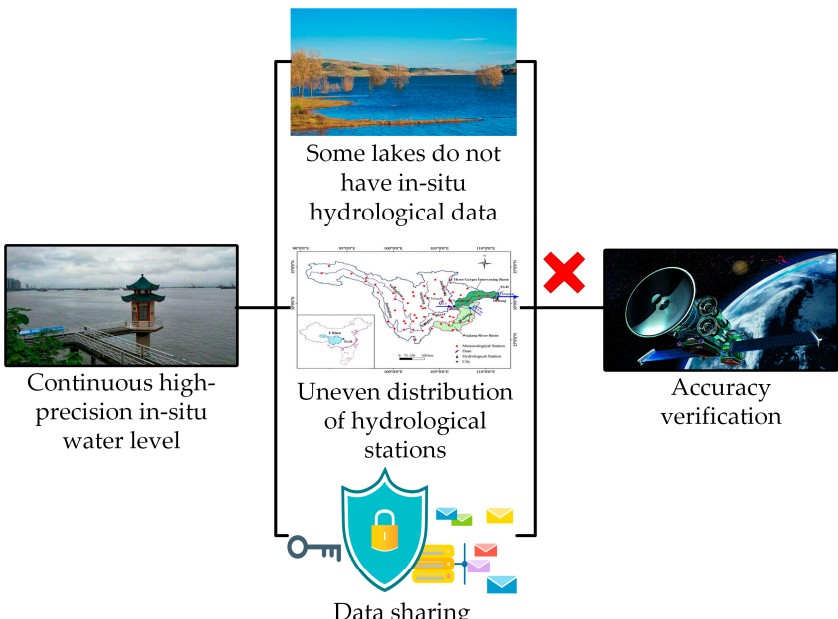

**Figure 6.** The problem of accuracy verification of altimeter satellite.

The challenge for applications and research services related to inland waters is the need for interdisciplinary knowledge in altimetry data [7]. Therefore, researchers need to understand the value of altimetry products. In the development stage of altimetry satellites, the latest development of airborne DEMs such as Jason-3 and SAR technology (CryoSat-2, Sentinel-3) or lidar (ICESat, ICESat-2) can help improve the measurement results. But it will be limited by the size of the running track and footprint, and it can only allow us to monitor the water level change information of a limited number of lakes and rivers. In the future stage of altimetry satellites, with the emergence of new altimetry satellites and technical methods, these studies will undergo great changes, such as the generalization of SAR and the evolution of wide-area interferometry (SWOT, WiSA). The SWOT mission will be an important step toward improving global coverage and high spatiotemporal resolution [156].

In addition to monitoring information on changes in water levels and volumes in inland waters, another challenge is how to derive river flows. Without the support of measured data, we cannot directly observe the required information from a spatial perspective. At present, many researchers have proposed algorithms [157,158] and model outputs [159]

that combine measured data with radar altimetry data. In addition, radar altimetry data are used for the calibration and evaluation of hydrological models [160–162]. In addition to the above, there are numerous studies using radar altimetry data to understand surface water storage and changes in rivers and floodplains [163,164].

### 3.2.2. Combining Deep Learning with Multi-Source Data

Satellite altimeter data provide long-term reliable observational information on the global ocean, inland waters, sea ice, etc., which makes it possible to understand changes in oceans, lakes, etc., from a dynamic perspective, especially in areas lacking measured data. In addition, near real-time data is available through altimeters, which drives many time-series-related applications (for example, flood forecasting in wet years or reservoir monitoring in dry periods). However, there are still many limitations and major challenges, such as limited coverage, temporal sampling, etc., if the altimeter data is to be widely used. As the constellation of altimetry satellites continues to expand, the quantity and quality of data and information products will increase. Both researchers and policymakers can gain more valuable information from it. In this way, the greater potential of altimetry data can be unlocked.

Research on the fusion of deep learning and multi-source data. The study found that the application of AI in the field of altimetry is still in the exploratory stage [6,143,165]. Large amounts of spatial data and integrated data (such as synthetic altimetry data and other available numerical models) can benefit from tools developed in the field of AI science. Through these tools, we can process, interpret, and understand this large amount of data, thereby increasing the value of such data. Complex models may not be able to perform full temporal or spatial analysis, while AI can simplify complex models. Therefore, AI can better consider the physical properties of the measurements in these analyses, or provide necessary auxiliary information [166].

When the data coincidence is low or even no coincidence, the monitoring results will be affected by the operating time and operating status of each altimetry satellite. We can use deep learning to study data fusion between multi-source altimetry satellites to solve the problem of missing data from hydrological stations and low data coincidence of multi-source altimetry satellites (Figure 7). The construction of a virtual hydrological station based on deep learning can not only predict the water level information but also provide a reference for the missing measured data. If there is a large gap between the predicted water level and the measured water level, the health status of the monitoring equipment can be inferred. When fusing the data between multiple altimetry satellites, using deep learning to build a data model of a single altimetry satellite can make the data segments of the altimetry satellites have a longer time coincidence, and the fusion can be more accurate.

Research on machine learning and processing methods of satellite altimetry signals. The use of machine learning techniques has made it possible to improve the performance of some signal processing techniques based on traditional methods and overcome their inherent limitations [167]. With the successful application of machine learning techniques in many engineering fields, the radar community has also begun to apply machine learning techniques to solve classic radar problems and deal with traditional problems from a new perspective [168]. Among them, the use of machine learning to optimize the waveform of radar signals will further improve the ability of radar detection, classification, identification, localization, and tracking [169]. At present, for the monitoring of small water areas, the "Land Pollution" in the radar waveform is the main source of error, and it has become possible to use machine learning to reduce or even eliminate "Land Pollution" [170].

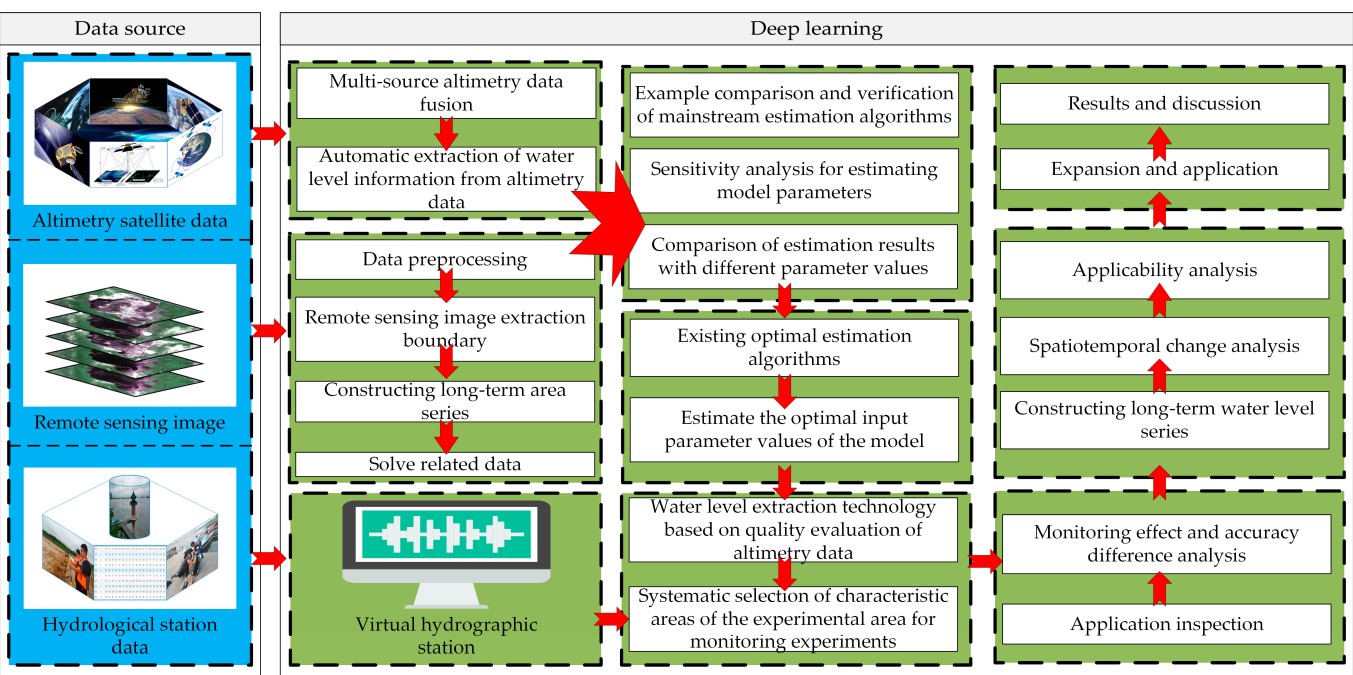

**Figure 7.** The whole process of inland lake monitoring based on deep learning.

### 3.2.3. Aggregation of Application Areas

Satellite altimetry has become an important technology for monitoring global ocean conditions. We can not only use satellite altimetry to analyze and forecast sea surface waves but also invert and estimate the sea surface wind speed field. It can also be used to monitor changes in sea level, and determine changes in ice surface height and the balance of ice sheet mass. Satellite altimetry data is an extremely important tool for monitoring ocean dynamic phenomena when studying the effects of atmospheric effects, marine meteorology, and ocean environmental characteristics on climate and their interactions. At the same time, satellite altimetry monitoring data is also a very important data source in the prediction of air-sea models. It can provide an analytical basis for the forecast of global hazard marine phenomena such as El Niño, La Niña, North Atlantic Oscillation, or Pacific Decadal Oscillation. This all benefits from its relatively dense spatial coverage, which can provide a dense network to monitor extreme events [163].

Many studies have mentioned that the daily water level of the upper reaches of the Yellow River can be estimated by satellite altimetry. This data can be used to predict the potential of downstream flows [171,172]. The Bangladesh Water Development Board (BWDB) also used Jason-2 altimetry data to make decisions on the flood season between July and October 2021, and carry out the protection of life and property safety across the country (www.ffwc.gov.bd (accessed on 6 June 2022)) (www.bwdb.gov.bd (accessed on 6 June 2022)). In the report materials of the "Summary of Ecological Protection and High-quality Development Planning of the Yellow River Basin" issued by the Central Committee of the Communist Party of China and the State Council in 2021 (http://www.gov.cn/zhengce/2021-10/08/content_5641438.htm (accessed on 6 June 2022)), it pointed out that "Using the Internet of things, satellite remote sensing, drones, and other technical means to strengthen the dynamic monitoring and scientific analysis of hydrology, meteorology, land disasters, rain conditions, flood conditions, drought conditions, and other conditions. We can build a comprehensive digital platform to achieve cross-regional and cross-departmental exchange and sharing of data resources. We define it as the "Smart Yellow River".

At the same time, we consider flood disasters as one of the worst natural disasters in the world. The population living in floodplains is increasing, and the probability of extreme

weather events has increased in recent years. All of these will cause life and property to be affected [173]. Although altimetry data enables monitoring of many watersheds, operational monitoring of some special waters (such as permanently or seasonally submerged vegetated plains) remains challenging [91]. For example, in the Amazon Basin, the biggest limitation to the widespread use of altimetry remains the relatively low time sampling [97–99]. In the Nile River Basin, the altimeter water level is highly consistent with the measured water level, but the estimated discharge from satellite data is inconsistent with the measured one, mainly due to overestimated inflow and uncertain outflow [174]. Due to the uncertainty of measurement [45,46], subtle changes inside the ice sheet are difficult to capture [42], and these deviations will interfere with subsequent studies, and the subsequent launch of ICESat-2 is eliminating these problems [40]. Further research is needed specifically.

Therefore, it is crucial to establish flood forecasting systems in flood-prone areas [175–178]. We can use multi-source data, including altimetry near real-time monitoring data, to build forecasting systems. It can assist in emergency preparedness and decision making to reduce the loss of life and property in flood events.

## 4. Conclusions

At present, there have been many successful cases of using satellite altimetry technology to monitor hydrological resources. Some researchers have begun to combine multi-source altimetry data, satellite gravity data, remote sensing satellite data, etc., to study surface water storage [6,74], and their corresponding climate changes, such as using multi-source altimetry data and remote sensing data to calculate changes in lake water storage, combined with GRACE data for lake water balance analysis and climate change response research of lakes on the Tibetan Plateau [10]. Changes in terrestrial water storage in the Lake Victoria basin can be studied using altimetry data, GRACE data, and rainfall data [76]. The contribution of the Antarctic ice sheet and the Greenland ice sheet to changes in global ocean quality can also be calculated and analyzed using altimetry data and GRACE/GRACE-FO data [12–18,179–181]. The changes in glacier mass in the Tibetan Plateau and the Indian Basin are also obtained by fusing multi-source data [77], for example, using data such as altimetry data, GRACE time-varying gravity field data, and the array for real-time geostrophic oceanography (Argo) ocean temperature and salinity data set, the global and Antarctic glacial isostatic adjustment (GIA) models can be used to analyze the global and effects of regional mass changes [78]. For ice thickness inversion and snow depth in the Arctic, the backscatter coefficient and echo waveform of the data can also be used for research [79,80,182–185].

This highlights that multi-source data, with its high-precision monitoring capability, has shown a certain potential in different research aspects. On the one hand, if the water level observation sequence data from satellite altimetry is combined with regional meteorological data, the application research on the evolution trend of lake waters under the condition of climate change can be realized [81]. On the other hand, using multi-source altimetry data and GRACE data to study lake water storage changes and lake water balance analysis in the Tibetan Plateau region verifies the feasibility and effectiveness of altimetry data for monitoring inland lakes, but the accuracy of altimetry data still needs further improvement. The contribution of glaciers to the lake water balance cannot be quantitatively estimated and further research is needed [186–189].

Although satellite altimetry observation techniques have been introduced into the remote sensing of hydrological resources, further studies are urgently needed to expand its benefits and applications. Therefore, this research aims to enhance the satellite altimetry observation techniques and methods for understanding, modeling, and predicting these phenomena and summarizes the research progress and existing problems of satellite altimetry technology in the field of inland lake monitoring. On this basis, we propose a multi-source altimetry data fusion method based on deep learning and strive to improve

the existing fusion algorithms. All the work hopes to provide some reference for the application and development of satellite altimetry technology in the future.

To sum up, the interdisciplinarity and the integration of satellite altimetry technology can provide effective technical support and direction expansion for the refined application research of surface water hydrology.

**Supplementary Materials:** The following supporting information can be downloaded at: https://www.mdpi.com/article/10.3390/rs14194904/s1, Figure S1: The Poseidon series altimeter of Jason satellite.

**Author Contributions:** Conceptualization, W.L. (Wei Li), X.X. and W.L. (Wanqiu Li); methodology, W.L. (Wei Li), and X.X.; formal analysis, W.L. (Wei Li), and X.X.; investigation, W.L. (Wei Li), and X.X.; resources, W.L. (Wei Li), and X.X.; data curation, W.L. (Wei Li), and X.X.; writing—original draft preparation, W.L. (Wei Li), and X.X.; writing—review and editing, W.L. (Wei Li), X.X., W.L. (Wanqiu Li), M.v.d.M., Y.H., X.L., and Q.W.; visualization, W.L. (Wei Li), X.X., and W.L. (Wanqiu Li); supervision, Q.W.; project administration, W.L. (Wei Li), and H.Y.; funding acquisition, W.L. (Wei Li), and H.Y. All authors have read and agreed to the published version of the manuscript.

**Funding:** This research was funded by the National Natural Science Foundation of China (41861061, 41930101, 42204006), the China Postdoctoral Science Foundation (2019M660091XB), the Key Laboratory of Geography and National Condition Monitoring, the Ministry of Natural Resources (2022NGCM01), the State Key Laboratory of Geo-Information Engineering and Key Laboratory of Surveying and Mapping Science and Geospatial Information Technology of MNR, CASM (2022-01-13), the Open Research Fund Program of the National Cryosphere Desert Data Center (E01Z790201/2021kf07), the Natural Science Foundation of Gansu Province (20JR10RA271, 21JR7RA317), the "Young Scientific and Technological Talents Lifting Project" Project of Gansu Province in 2020 (Li Wei), "Tianyou Youth Lifting Project" Program of Lanzhou Jiaotong University (Li Wei), and the Innovation and Entrepreneurship Education Reform and Cultivation Project in Gansu Province (1A50190117).

**Acknowledgments:** We also thank the editors and three reviewers for their constructive feedback, which improved the readability and reasoning of the paper. We thank AVISO (https://www.aviso.altimetry.fr/ (accessed on 6 June 2022)) for the altimetry dataset. Thanks to UMCES (https://ian.umces.edu/media-library/ (accessed on 6 June 2022)) and pngimg (https://pngimg.com/ (accessed on 1 August 2022)) for providing image materials. Thanks to VOSviewer software for helping us draw a knowledge map.

**Conflicts of Interest:** The authors declare no conflict of interest.

## Abbreviations

The following abbreviations are used in this manuscript:

| | |
|---|---|
| ATSAT | A MATLAB-based software for multi-satellite altimetry data analysis |
| AVISO | CNES data center for Altimetry and DORIS products |
| BRAT | Basic Radar Altimetry Toolbox |
| CAST | China Academy of Space Technology |
| CNES | Centre National d'Etudes Spatiales |
| CNSA | China National Space Administration |
| COSDSC | China Ocean Satellite Data Service Center |
| CSA | Canadian Space Agency |
| CTOH | Center for Topographic studies of the Ocean and Hydrosphere |
| DAHITI | Database for Hydrological Time Series over Inland Waters |
| DGFI-TUM | Deutsches Geodätisches Forschungsinstitut (Deutsches Geodätisches Forschungsinstitut is a research institute of the Technical University of Munich (TUM)) |
| ESA | European Space Agency |
| EU | European Union |
| NASA | National Aeronautics and Space Administration |
| GFO | GEOSat Follow-on |

| | |
|---|---|
| G-REALM | Global Reservoirs and Lakes Monitor |
| ISRO | Indian Space Research Organisation |
| LEGOS | Laboratory of Space Geophysical and Oceanographic Studies |
| ncBrowse | A Graphical netCDF File Browser |
| NCO | NetCDF Operator |
| NOAA | National Oceanic and Atmospheric Administration |
| NSIDC | National Snow and Ice Data Center |
| NSOAS | National Satellite Ocean Application Service |
| OpenADB | Open Altimeter Database |
| RADS | Radar Altimeter Database System |
| SASWE | Sustainability, Satellites, Water, And Environment |
| UKSA | UK Space Agency |
| U.S.Navy | United States Navy |
| USDA | U.S. Department of Agriculture |

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
