# Peer review of "Monitoring of Hydrological Resources in Surface Water Change by Satellite Altimetry"

_remotesensing, doi:10.3390/rs14194904_

Round 1

Reviewer 1 Report

This paper describes the monitoring of hydrological resources by satellite altimetry, as in its title. Various changes and improvements are necessary, as follows.

1. A detailed check of syntax and grammar in the whole text. Please avoid writing too long sentences and too long paragraphs (e.g. page 3).

2. In the Introduction section, the content of text and figures are summarized, while the first and second stage development representations are missing in Figure 3. The development of the whole process of satellite altimetry can be listed and displayed in a table.

4. In the Research Question content. I suggest displaying the information of the existing data sources, listing them in the form of tables, and giving the connection and publishing organization. In order to have a lot of information, it is convenient to display the content and enhance the value of the paper

The existing programs for data processing can be listed. And give a detailed explanation and analysis, respectively, of what programs can be processed. For example, it can be used for the study of spatial and temporal resolution (What's the disadvantage). Please add references, or shortly explain.

5. In the Challenge and opportunity. It is recommended to list the advantages of artificial intelligence/machine learning/satellite altimetry data processing in time series, which can make up for the advantages of existing programs or data.

In this way, the application advantages of the machine can be clearly introduced. For example, it can improve intelligence, reduce interaction, and improve spatial resolution. These can be used to study hydrological resource monitoring in local watersheds.

6. In the Conclusion and Outlook section, please reform the sentences to emphasize more on the meaning of the conclusions. The research on hydrological resources of satellite altimetry in a specific area should be sorted out, and a brief outlook should be given.

7. There is a reference citation error (line 313, page 8).

Author Response

Thanks for your systemic review. We agree with this suggestion and have modified the terminology throughout the text as appropriate. Firstly, to sort out the logical framework, we have arranged the full-text catalog. This form of presentation will allow more information in the paper. We also give some clarifications and suggestions on the relevant data. We greatly appreciate your consideration of our revised manuscript, and we look forward to receiving comments from you.

Reviewer 2 Report

This work is interesting and timely to the satellite altimetry community, it's worth publish in remote sensing. but the current manuscript need further improved, mainly lies in:

1.please add more literature in the first paragraph[page 1 31~34], and the first paragraph is a little  short, it is recommended to add a little more to introduce the topic. Besides,  recently review work on Satellite Altimetry should be noticed: 

Yang, L., Lin, L., Fan, L., Liu, N., Huang, L., Xu, Y., Mertikas, S.P., Jia, Y. and Lin, M., 2022. Satellite Altimetry: Achievements and Future Trends by a Scientometrics Analysis. Remote Sensing, 14(14), p.3332.

2. Page 2.......... changes in inland lakes 53 and other waters [2]:better to cite some recently references

3.Subtitles such as 5.2.1~5.2.3 need to be revised and closely related to the theme, e.g. our topic is hydrological resources by satellite altimetry, better to modify to : Application of Machine learning in...? the current sub-title is a little simple

4.Page 8, line 313: with error in citation. Be carefule

5.it will be better to display the information of the existing data sources (e.g. Figure 1 and the Research Question content) in tables, and give the connection and publishing organization, it is convenient to display the content and enhance the value of the paper.

6.In the Conclusion and Outlook section. The research on hydrological resources of satellite altimetry in a specific area should be sorted out. For example: Amazon, Yangtze River, Yellow River, Heihe, etc. The research results of many well-known regions should be listed, and their unsolved problems can be briefly prospected.

Author Response

Thanks for your systemic review. We agree with your suggestions and comments. The article has been revised. Firstly, to sort out the logical framework, we have arranged the full-text catalog. This form of presentation will allow more information in the paper. We also give some clarifications and suggestions on the relevant data. We greatly appreciate your consideration of our revised manuscript, and we look forward to receiving comments from you.

Reviewer 3 Report

Review comments

Review of the manuscript titled “Monitoring of hydrological resources in small-scale river basins by satellite altimetry: A review and perspectives”

Major comments

1.     The title is quite attractive however, the manuscript does not cover the titled research. The study is limited to inland lakes. The authors are suggested to extend the study to cover the river basins hydrology focusing on water resources both in liquid and solid form. Without the resolution of this comment, the study is too limited.

2.     The authors jump from TOPEX/Poseidon mission in 1992 to the most recent decade (2010-2020). As a review paper, the authors have to cover all the altimetry missions. There has been a lot which the authors missed in the time missed by the authors e.g. GRACE, ICESat-1 and ICESat-2. The authors need to significantly improve the literature review/introduction part of the manuscript. The authors have suddenly indicated ICESat in section 4 without any prior background. The authors need significant improvement in the literature review, and should broaden the research focus.

3.     The authors mostly limit their literature to water level, sea level and lakes. However, frozen water which is a major component of the hydrology is completely missed. For glaciers/snow, ICESat data has been widely used from the Polar Regions to high mountains worldwide (Kääb et al., 2012, 2015; Zwally et al., 2005, 2008, 2011a, 2011b). The authors are more concentrating on inland lakes “The study of inland lakes and reservoirs has received increasing attention due to increasing population, increasing water demands, and changing climate” in contrast to their primary focus (hydrological resources of river basins). The authors need to improve their research and cover the primary focus area.

Minor comments

·       The statement “During this period, satellite radar altimeter data has been successfully applied to fish tracking [8], severe storm forecasting [9], oil spill response [10], ship route tracking [11], iceberg monitoring [12], marine wildlife habitat monitoring [13], wetland dynamics monitoring [14], reservoir lake monitoring [15], flood forecasting [16], monitoring of changes in river levels [17, 18] and the development of offshore wind farms [19], among which fish tracking and the development of offshore wind farms, are already commercially available.” Is incomplete, the authors have to cover all the research/application areas (as also mentioned in comment 2).

·       In the statement “In response to these problems, the use of satellite altimetry technology in reservoir operations and river system modeling has been successfully carried out in operational environmental monitoring of transboundary basins, such as the Mekong River Basin [21], the Ganges-Brahmaputra in Southeast Asia River Mekhna Valley [15], Indus Valley [22], Nile Valley [23]”, the research on the Indus is studied with satellite altimetry (as mentioned in major comment 2). The authors may refer to the most relevant research.

·       The altimetry data has been widely used for water resources and frozen land surface. Also the data from satellites have been validated on the ground even in high altitudes >6000 m a.s.l. altitudes (Muhammad and Tian 2020 - Mass balance and a glacier surge of Guliya ice cap in the western Kunlun Shan between 2005 and 2015), therefore the statement needs revision “one of the main challenges remains the need to demonstrate the reliability of radar altimetry data”.

·       Line 198-204: I also disagree, relevant agencies of different constellations provide support in terms of software for data processing. So, it does not seems a challenge. Similarly, these agencies offer open trainings and share materials of training for data processing. So the points mentioned here are invalid. Also, the authors negate in the 205 sentence, what they say in the previous paragraph.

References

Kääb A, Berthier E, Nuth C, Gardelle J, Arnaud Y. Contrasting patterns of early twenty-first-century glacier mass change in the Himalayas. Nature. 2012; 488:495–498. 

Kääb A, Treichler D, Nuth C, Berthier E. Brief Communication: Contending estimates of 2003-2008 glacier mass balance over the Pamir–Karakoram–Himalaya. The Cryosphere. 2015;9:557–564

Muhammad, S., Tian, L., & Khan, A. (2019). Early twenty-first century glacier mass losses in the Indus Basin constrained by density assumptions. Journal of Hydrology, 574, 467-475.

Zwally, HJ and Giovinetto, MB (2011a) Overview and assessment of Antarctic ice-sheet mass balance estimates: 1992–2009. Surv. Geophys., 32, 351

Zwally, HJ and 11 others (2011b) Greenland ice sheet mass balance: distribution of increased mass loss with climate warming. J. Glaciol., 57(201), 88–102 (doi: 10.3189/002214311795306682)

Zwally, HJ and 7 others (2005) Mass changes of the Greenland and Antarctic ice sheets and shelves and contributions to sea-level rise: 1992–2002. J. Glaciol., 51(175), 509–527 

Zwally, HJ, Yi, D, Kwok, R and Zhao, Y (2008) ICESat measurements of sea ice freeboard and estimates of sea ice thickness in the Weddell Sea. J. Geophys. Res., 113(C2), C02S15

Author Response

Thanks for your systemic review. We agree with your suggestions and comments. The article has been revised. Firstly, to sort out the logical framework, we have arranged the full-text catalog. This form of presentation will allow more information in the paper. We also give some clarifications and suggestions on the relevant data and program. And the application types and regions are divided. Among the existing limitations and opportunities, deep learning methods are integrated into the solution of satellite altimetry data processing. We greatly appreciate your consideration of our revised manuscript, and we look forward to receiving comments from you.

Round 2

Reviewer 2 Report

The author's newly submitted version has been greatly improved, and he has responded to my questions one by one in the last round, the work can be published after minor revision, mainly in the figures and literature citations

1. In Figure 1 : it is better to change the picture of the sun, to be more rigorous.

2.Page 2 line 62: The citations is not very standardized, it is recommended to modify as follows:

11-21, to make it easy to follow and better looking.

Check the full text of the literature annotations and unify them.

3.Figure 2 : if possible, Provide higher resolution figure

4. Figure 4: A white background could be better

Author Response

Thank you very much to the Editor and Reviewers for our manuscript's constructive comments and suggestions. We have updated some content in the revised manuscript.

Thanks for your suggestion.

We modified Figure 1 (Line 49, Section 1. Introduction) in the revised manuscript.

We modified the content in the revised manuscript:

Line 62, Section 1. Introduction: evapotranspiration [10], snow and ice [11-21], are measurable using remote sensing at various spatial-temporal resolutions and accuracies [7].

Line 117, Section 2.1.2. Hydrological in liquid form: The application of satellite altimetry technology has expanded from initial sea-level monitoring to monitoring water level changes in inland lakes and other waters [73-82].

Line 470, Section 4. Summary: The contribution of the Antarctic ice sheet and the Greenland ice sheet to changes in global ocean quality can also be calculated and analyzed using altimetry data and GRACE/GRACE-FO data [12-18,181,182,183].

We modified Figure 2. (Line 85, Section 1. Introduction) in the revised manuscript.

We modified Figure 4 (Line 150, 2.2.1. Section 2.2.1 Availability of data) in the revised manuscript.

Reviewer 3 Report

Review comments

Review of “Monitoring of hydrological resources in surface water change by satellite altimetry: A review and perspectives”

The authors are appreciated for significant improvement of the manuscript. I have few more comments before acceptance.

Comments

Some of my comments are not properly answered. I am putting parts of the comments again for proper response.

1.      In my first comments, many studies with altimetry data applications in the Indus and surroundings are not proper 

2.      The authors are advised to revise “the mass balance of ice sheet” and “the mass balance of ice sheet and mountain glaciers

Author Response

Thanks for the suggestion. We have reviewed the corresponding research to illustrate the results of satellite altimetry studies in the Indus Valley, and updated the relevant references to these areas. Line 137-142, Section 2.1.2. Hydrological in liquid form.

We found descriptions in the manuscript for both definitions. Mountain glaciers are not involved in our study. To avoid ambiguity, we removed the content (Line 621, Section 4. Summary) in the revised manuscript
